# Potential Benefits of Behaviors and Lifestyle for Human Health and Well-Being

**DOI:** 10.3390/nu17203253

**Published:** 2025-10-16

**Authors:** Łukasz Stachera, Karolina Góras, Klaudia Janowska, Erwina Muszkat-Pośpiech, Anna Wojciechowska, Grażyna Świderska-Kołacz, Szymon Zmorzyński, Joanna Czerwik-Marcinkowska

**Affiliations:** 1Institute of Biology, Jan Kochanowski University, 25-406 Kielce, Polandgrazyna.swiderska-kolacz@ujk.edu.pl (G.Ś.-K.); joanna.czerwik-marcinkowska@ujk.edu.pl (J.C.-M.); 2LO I Technikum TEB Edukacja, ul. Puławskiego 25, 40-001 Katowice, Poland; erwina.muszkat@teb.edu.pl; 3Department of Geobotany and Landscape Planning, Nicolaus Copernicus University, 87-100 Toruń, Poland; ankawoj@umk.pl; 4Institute of Human Sciences, Academy of Zamość, 22-400 Zamość, Poland; szymon.zmorzynski@akademiazamojska.edu.pl

**Keywords:** functional food, health benefits, lifestyle

## Abstract

Background/Objectives: Proper nutrition and a balanced lifestyle are key determinants of overall human well-being, influencing both physical and mental health. Likewise, physical activity and daily lifestyle choices play a vital role in sustaining the proper functioning of physiological systems and preventing chronic diseases. Methods: This qualitative study was conducted between April and July 2025 among students and employees of the Faculty of Exact and Natural Sciences at Jan Kochanowski University. Data were gathered using the standardized KomPAN^®^ 2.2 questionnaire, which assessed dietary habits, lifestyle behaviors, and demographic factors. Participants were selected according to defined inclusion and exclusion criteria, focusing on full-time students and research employees reporting well-being-related difficulties. Statistical analysis employed multivariate techniques, including Indicator Value Analysis in PAST 5.2.1 and Principal Component Analysis in Canoco 5.0, to identify significant differences between groups. Results: The results showed that students consumed more fruits and vegetables but also more fast food and sweetened beverages, while employees differed mainly in lifestyle characteristics such as urban residence and higher education level. Gender-related analyses indicated that women selected specific food products more often, whereas men were more physically active. Conclusions: The findings highlight the need for targeted well-being and nutrition support programs within academic institutions.

## 1. Introduction

The health and well-being of human and of populations are the result of three groups of factors: genetics, environment and behavior. Only the last is mostly dependent on the choices of individuals, but assessment, interventions and tailored changes are possible [1,2]. The World Health Organization (WHO) defines health as a state of complete physical, mental, and social well-being [3]. Diener [3] stated that subjective well-being refers to a person’s cognitive and affective evaluations of his or her life. Human well-being is closely associated with work, family, health, and economic benefits. Health-related behaviors may be divided into two categories: those that promote health, commonly referred to as health-promoting behaviors such as rational nutrition, physical activity, and stress management, and those that pose a threat to health, so-called health-risk behaviors, including smoking, alcohol consumption, the use of psychoactive substances, or the misuse of medications [4,5]. Lifestyle is expressed by daily work and leisure profiles, including activities, attitudes, interests, opinions, values, and allocation of income [6]. From a psychological point of view lifestyle derives from people’s self-image or self-concept (the way they see themselves and believe they are seen by the others), including self-esteem and self-efficacy. Lifestyle is a composite of motivations, needs, and wants and is influenced by factors such as culture, family, reference groups, and social class. Among university students, well-being is of particular importance, as it constitutes a key factor in achieving satisfaction; however, it is simultaneously at risk, as students encounter numerous academic, social, emotional, and psychological challenges [7,8,9]. Douwes et al. [10] observed that despite the widespread consensus on the importance of student well-being, a clear definition continues to be lacking. A major recurring theme was well-being as a balance in the interplay between efforts directed towards studies and life beyond studies. Their study contributed to the body of knowledge on the well-being of students in higher education and provided suggestions for educational institutions, such as incorporating a holistic perspective on students and learning; and focus points for the development of policies and practices.

Rational nutrition involves the intake of nutrients that exert beneficial effects on health. Healthy eating is one of the primary determinants of both quality and longevity of life. A proper diet supports the maintenance of homeostasis, contributes to the optimal functioning of organs and organ systems, influences external appearance (e.g., skin, hair, nails), and affects both physical and mental fitness [11,12]. Energy requirements, however, depend on lifestyle, daily physical activity, age, health status, and sex [13,14]. Poor dietary habits may contribute to the development of numerous diseases, including overweight and obesity, as well as diabetes, atherosclerosis, myocardial infarction, and cancer [13]. Rational nutrition further encompasses appropriate food selection, culinary processing, portion size, and meal frequency and distribution throughout the day. Eating habits established during youth exert a profound influence on dietary patterns and health status in older populations [13,14]. Researchers [15,16] agree on the notion that nutritional factors have major impact on the risk of age-associated chronic non-communicable diseases and mortality. Dominguez et al. [15] described some specific dietary patterns with evidence of associations with reduction in the incidence of chronic diseases allowing older adults to live a long-lasting and healthier life and confirming the powerful impact nutrition can exert on healthy aging.

A crucial element of health-promoting behavior is physical activity. It is well known that sport is a multifaceted concept, encompassing the type of discipline, goals, training approaches, and attitudes toward competition. Numerous researchers highlight the benefits of physical activity [17,18,19,20,21]. Moderate exercise prevents cardiovascular diseases, and facilitates reduction in cholesterol levels and stabilization of lipid metabolism, thereby reducing the risk of atherosclerosis and lowering blood pressure [18,19]. Regular physical activity also alleviates symptoms of depression and anxiety, improves mental performance and well-being through elevated endorphin levels, and counterbalances the strain of intellectual work, enhancing resilience to stress and fatigue. Moreover, systematic physical activity contributes positively to human cognitive functioning [22].

It is also noteworthy that cigarette smoking represents one of the most widespread addictions worldwide. Nicotine dependence remains a major social, economic, and health concern. Chronic smoking, similar to the use of other addictive substances, exerts harmful effects on the human body [23]. The toxic properties of cigarette smoke led to widespread pathological changes across multiple organs and tissues. In both smokers and non-smokers, nicotine triggers the release of adrenaline and noradrenaline, resulting in increased blood pressure, elevated heart rate, enhanced myocardial contractility, and vasoconstriction [24,25,26,27,28]. There is evidence that behavioral tobacco use is adopted because close friends or family members have developed this habit [29,30]. Saari et al. [29] stated that the impact of a smoker as a close friend is greater than that of a smoking parent or sibling in school age when it comes to smoking behavior in adulthood.

While this study primarily focuses on lifestyle factors such as dietary habits, physical activity, and selected health-related behaviors, it is important to note that psychological variables and broader aspects of well-being are discussed only theoretically. The measurement tool employed in this research, the KomPAN^®^ questionnaire [31], does not comprehensively assess all psychological dimensions of well-being, such as stress levels, self-efficacy, or emotional resilience. This represents a limitation of the current study, as the complexity of human well-being extends beyond observable behaviors to include cognitive and emotional processes. Therefore, future research aiming to capture a more complete picture of well-being may require the inclusion of additional validated psychological instruments, enabling the simultaneous assessment of both behavioral and psychological determinants of health and well-being.

The aim of this study was to investigate differences in lifestyle patterns and dietary preferences, first, between university employees and students, and second, between women and men. Despite numerous studies exploring nutrition and lifestyle determinants of well-being, there remains a lack of comparative analyses simultaneously examining dietary and behavioral patterns among different academic groups within the same institutional context. Previous research has largely focused either on students or on employees, often treating these populations separately. However, universities constitute complex micro-environments where both groups coexist, interact, and may influence one another’s well-being and health-related choices. Therefore, conducting a comparative analysis within this shared context offers a valuable opportunity to identify both convergences and divergences in behaviors that may affect long-term health outcomes. This study provides a novel contribution jointly examining dietary, lifestyle, and demographic factors to identify differences in daily habits and well-being-related behaviors between university students and academic employees. By additionally accounting for sex-based variations, the research deepens understanding of how gender interacts with occupational and educational contexts in shaping both health-promoting and risk-related behaviors. The findings are expected to inform the development of targeted well-being initiatives, nutritional education programs, and health promotion strategies tailored to the specific needs of university populations.

It was hypothesized that significant differences exist in lifestyle patterns and dietary preferences between university employees and students, and between gender groups, reflecting variations in socioeconomic status, daily routines, and health awareness.

## 2. Materials and Methods

### 2.1. Study Design

A qualitative study was conducted at the Jan Kochanowski University in 2025 among employees and students of the Faculty of Exact and Natural Sciences. This Faculty offers approximately 19 different associate degrees, bachelor’s, and master’s programmes, ranging from biology, chemistry, physics studies to geography and environmental sciences, and mathematics. In total, over 110 students study at this institution. Faculty is a leader in the number of scientific and research projects and published publications and that’s why it was decided to conduct the study at this faculty due to the fact that employees (teaching staff) and students publish most effectively and most often participate in international conferences.

### 2.2. Procedures

Data were collected through the diagnostic survey method using a questionnaire technique interview between April and July 2025. The measurement tool was the KomPAN questionnaire, version 2.2 (KomPAN® Questionnaire for the study of opinions and dietary habits for adolescents (16–18 years) and adults version 2.2—self-administered questionnaire) attached in Appendix A. The KomPAN® questionnaire was developed and validated for the adult Polish population. Although this tool had previously been used in studies among students, no separate validation was conducted for this group, which should be considered a potential limitation of this study.

The questions addressed the frequency of consumption of specific food product groups as well as selected aspects of respondents’ lifestyles. The survey additionally collected basic demographic information, along with data reflecting the health status and economic conditions of the participants.

Employees and students were informed about the study by a message on the “Wirtualna uczelnia”, which is available to students and employees. The purpose of the study was explained and a link to a participation form was provided in the message. Simultaneously, the Head of Dean of the faculty were requested to advertise the same information and opportunity to participate, among students at all institutes. A total of 39 students and 20 employees (teaching staff) signed up for a questionnaire and completed it.

### 2.3. Sample Selection

Certain selection criteria were set to select information rich cases that would help to obtain that understanding. For the selection of participants from those who signed up, the following criteria were used: (a) participants had to be a fulltime student and should have had experience of self-reported well-being problem(s) during their studies, (b) participants had to be a fulltime research workers and should have had experience of self-reported well-being problem(s) during their works (Table 1).

### 2.4. Ethical Considerations

The study was conducted in accordance with the Declaration of Helsinki and approved by the Bioethics Committee of the Academy of Zamość (No. KBAZ/3U/2024). Each participant received information about study purposes and was provided with instructions for questionnaires completion. The participation in the study was voluntary. The results obtained were anonymized and used only for research purposes. The participants were not known to each other beforehand. Our study used an anonymous paper-based questionnaire. We did not use signed consent forms because this would have contradicted our guarantee of anonymity to the participants. Instead, we followed a standard method for this type of research. An information statement was printed at the top of each questionnaire, informing participants that the study was voluntary, anonymous, and explaining its purpose. We considered the act of completing the survey as consent to participate in the study.

### 2.5. Data Analysis

The dataset was statistically analyzed to obtain information on differences in life-style and food preferences between university employees and students, and between women and men, respectively. Basic statistical analysis was performed for all survey question results. Means and standard deviations were calculated. Because the data were not normal, the median was also calculated and Mann–Whitney’s tests were performed to identify questions with significant differences between the study groups. Basic statistics served as a prelude to multivariate analyses. These were performed in Statistica 9.0 version 9.0 [32].

Two multivariate techniques were used. The first involved analysis of indicator values, the purpose of which was to determine which of the questions statistically significantly indicated one of the study groups. The indicator value (IndVal) is expressed as a percentage, and its color-coded graphical representation is presented in the resulting graphs. The statistical significance of the indicator values is estimated based on a permutation test performed during the analysis. The analysis of indicator values was conducted using PAST 5.2.1 [33]. The second analysis was an indirect ordination technique, Principal Component Analysis (PCA). This technique identified the tendencies of the study groups to indicate specific responses. In the ordination diagrams, the questions that were statistically significant in the first analysis are marked in red. PCA was performed using Canoco 5.0 version 5.0 [34].

Basic statistical analysis was also supported by the Glass biserial correlation coefficient, which was used to assess the magnitude of the studied effect and the power of the nonparametric test [35].

## 3. Results

### 3.1. Sample Characteristics

Table 2 illustrates respondent characteristics dietary habits of students and employees (male, female, total). Twenty faculty employees aged between twenty-two to sixty-five years and thirty-nine students aged nineteen to twenty-eight years participated. Respondents came from five different institutes. Of this group, sixteen students consumed three meals per day, thirteen consumed four meals per day, eleven employees consumed three meals per day and four employees consumed four meals per day.

Table 3 provides a detailed breakdown of participant responses to the questionnaire. The first part of the table details the consumption frequency of various food products and beverages. Table 4 presents data related to lifestyle and personal characteristics, including the frequency of eating out, daily screen time, sleep duration, nicotine use, and self-assessed health status, nutritional knowledge, and physical activity. All respondents were categorized by group (Students, Employees) and subgroup (Male, Female).

### 3.2. Employees vs. Students

Surveys in these groups indicated that students eat significantly more vegetables and fruit (Q20, IndVal = 56.8, *p*-value = 0.012). At the same time, they also eat significantly more fast food (Q21, IndVal = 53.7, *p*-value = 0.042 and Q26, IndVal = 56.8, *p*-value = 0.032). Students also drink sweetened beverages (Q51, IndVal = 100, *p*-value = 0.004) and energy drinks (Q52, IndVal = 58.6, *p*-value = 0.023) significantly more often. However, students also spend significantly more time sleeping on weekends (Q88, IndVal = 53.4, *p*-value = 0.038).

In contrast, the only questions in which employees significantly differed were related to their lifestyle. Employees were significantly more likely to come from cities (Q104, IndVal = 60.4, *p*-value = 0.004) and have higher education (Q110, IndVal = 54.1, *p*-value = 0.0001). In this group of questions, Q105 was a statistically significant indicator for students—their households had more people, including minors (Q106) (IndVal = 65.0, *p*-value = 0.0001) (Figure 1).

These relationships were presented graphically in the ordination analysis (Figure 2). The diagram indicates a different approach to nutrition in the two study groups. In their responses, only four employees provided answers similar to the student group. The responses of both groups were essentially different.

Multivariate tests generally confirmed the information obtained using the Mann–Whitney test. However, they were not fully convergent. Among the questions for which the significance of differences between the responses of both groups was not confirmed were: 21 (rg = −0.16, U = 328, *p* < 0.05), 52 (rg = −0.28, U = 281, *p* < 0.05), 88 (rg = −0.29, U = 276, *p* < 0.05), and 106 (rg = −0.27, U = 284, *p* < 0.05). Furthermore, the Mann–Whitney test indicated significant differences between responses to questions 103 (rg = −0.74, U = 67, *p* > 0.001) and 109 (rg = −0.44, U = 220, *p* > 0.01), but this was not confirmed by the multivariate IndVal test. For questions with significant response differences, the power of the tests ranged from moderate (the lowest value was −0.36 for question 20) to very strong (the highest value was −0.74 for question 103). Detailed information on the results and power of the tests is provided in Appendix A.

### 3.3. Women vs. Men

The survey results revealed that women ate significantly more frequently during the day (Q19, IndVal = 55.3, *p*-value = 0.032). Women also reported significantly higher consumption of foods such as white rice, regular pasta, small-grain groats (Q24, IndVal = 55.9, *p*-value = 0.028), cottage cheeses (Q33, IndVal = 58.7, *p*-value = 0.001), and white meat (Q37, IndVal = 54.6, *p*-value = 0.043). The most pronounced differences between sexes, however, were related to lifestyle. Men were significantly more likely to engage in physical activity both at work (Q90, IndVal = 58.3, *p*-value = 0.009) and outside of work (Q91, IndVal = 55.9, *p*-value = 0.034), which suggests that women were more often employed in sedentary occupations. Additionally, as expected, men were significantly taller (Q96, IndVal = 62.2, *p*-value = 0.0001) and heavier (Q97, IndVal = 57.4, *p*-value = 0.003) (Figure 3).

These relationships were further visualized through ordination analysis (Figure 4). In contrast to the previous PCA, the separation between male and female groups was less distinct, indicating that sex-related differences were not as clearly manifested in this analysis.

Multivariate tests and the Mann–Whitney test in this case showed almost identical results. Significance of differences between the two groups’ responses was not confirmed only for questions 37 (rg = 0.28, U = 183, *p* < 0.05) and 91 (rg = −0.32, U = 172, *p* < 0.05). For the questions for which significant differences were found, the power of the tests ranged from moderate (the lowest value was 0.39 for question 24) to very strong (the highest value was −0.76 for question 97). Detailed information regarding the results and power of the tests is provided in Appendix A.

## 4. Discussion

Well-being of students and employees in university education is an important topic in both educational practice and research. In our study, the point was on the student and faculty employees perspective on well-being in university education, specifically regarding (a) how they define well-being and (b) what they consider as influencing factors. This study revealed numerous irregularities in the frequency of consumption of selected food products among both employees and students. All respondents reported intake levels below nutritional recommendations with respect to meal frequency, consumption of vegetables and fruits, and fermented dairy beverages. Deviations from the healthy eating model were also observed in the form of high consumption of sweetened carbonated drinks, fast food, and confectionery products. Individuals with higher education were more often interested in healthy nutrition and demonstrated greater knowledge in this domain. Higher educational attainment was frequently associated with regular meal consumption, avoidance of snacking between meals, and a preference for healthier food products. Conversely, those with lower education levels tend to consume fast food, sweetened beverages, and processed foods more frequently. Among women, higher education was also linked to greater nutritional knowledge.

According to the World Health Assembly (WHA), cereal products should be included in every meal of the day, as they provide an excellent source of complex carbohydrates, plant-based protein, and B vitamins. Vegetables and fruits should be consumed three to four times daily, supplying vitamin C, minerals, and dietary fiber. Milk and dairy products should also appear in three to four daily meals, serving as key sources of calcium, protein, and vitamins B2, A, and D. Meat, poultry, fish, and eggs are recommended in no more than two to three servings daily, given their high content of iron, protein, and B-group vitamins. Fat intake should be moderate, with a preference for plant-derived fats. Insufficient intake of essential nutrients contributes to a wide range of metabolic disorders and deficiency-related diseases, while excessive intake of macronutrients, particularly fats and simple carbohydrates, leads to weight gain and the development of diet-related diseases [36,37]. The dietary habits of Poles, particularly students but also academic staff in our study, frequently diverge from global recommendations for healthy eating. Nutritional errors are widely recognized yet commonly disregarded, stemming from economic constraints as well as limited knowledge of proper dietary principles [38].

Fermented milk beverages are low in calories, and their lactic acid bacteria have beneficial effects on gastrointestinal function [39,40]. In recent years, considerable attention has been given to the role of microorganisms colonizing the human body in maintaining health [41]. The intestinal microbiota is often referred to as a “newly discovered organ.” The composition of the gut microbiome is dynamic and influenced by diet, infections, stress, hygiene, and medication use (including antibiotics). Qualitative and quantitative changes in intestinal microbiota have been associated with the pathogenesis of diseases such as atopy, allergy, inflammatory bowel disease, irritable bowel syndrome, diabetes, celiac disease, and obesity. Dysbiosis and disturbances in the gut–brain axis are considered key factors in the pathophysiology of irritable bowel syndrome. The gut microbiota communicates with the host via intestinal epithelial, immune, and nervous system cells by producing and releasing molecules that transmit signals to distant organs. It is currently estimated that a significant proportion of signaling molecules circulating in mammalian blood originate from gut microorganisms [42,43,44]. Increasing attention has also been directed toward the impact of microbiota on the central nervous system (CNS). Numerous studies describe bidirectional communication between the gut and the CNS, the so-called “gut–brain axis.” Proper functioning of this axis is regarded as essential for gastrointestinal physiology. Thus, the terms microbiota and gut–brain axis are now used conjointly, often described as the “gut–brain–microbiota axis” [45]. Even normal brain responses to afferent stimuli may be modulated by altered gut microbial composition [46,47]. Evidence also suggests that the gut microbiota can be influenced by signals from the CNS and, in turn, affect brain function [48,49].

Dietary behavior models differ between women and men in both the quantity and quality of diet, as well as in meal frequency, timing, location, and accompanying health-promoting behaviors. Men are more likely than women to associate a healthy lifestyle with incorporating regular physical activity into daily routines rather than with dietary modification. Thus, physical activity may serve as an initial step toward improved eating habits, reflected in reduced caloric intake and improved nutritional quality of meals. Individuals engaged in sport are generally more attentive to their dietary profile, striving to balance macro- and micronutrient intake to enhance endurance performance. Women, on the other hand, tend to place greater trust in the benefits of healthy eating, exhibit stronger commitment to weight control, and make healthier food choices compared to men. Women also show a greater tendency to eat in social contexts, including snacking when others are eating or in response to stress. Moreover, women often prioritize the pleasure of consuming sweet foods over concerns about weight control or potential negative health outcomes. Men, by contrast, tend to prefer high-fat, strongly flavored foods and often consume sweet snacks discreetly, for example, while watching television. Men are also more likely than women to use dietary supplements and frequent fast food restaurants. Additionally, men more often report engaging in regular physical activity and modifying their diet for weight loss.

The perception of sex differences in food acquisition, preparation, and consumption has deep cultural and evolutionary roots. In pre-Neolithic societies, women were primarily gatherers, while men were hunters. Contemporary men may have inherited from hunters a stronger tendency toward high-energy, high-fat foods, whereas women may have inherited a preference for carbohydrates, combined with a socially reinforced emphasis on maintaining proper body weight, which contributes to their more frequent use of weight-reduction diets. These multiple factors influencing nutritional behavior provide the basis for distinguishing between male and female dietary strategies [7].

Physiological differences also shape dietary needs and health outcomes. Men have greater energy requirements due to genetically determined higher body height and mass. Women, by contrast, generally have a higher body fat percentage, and obesity is more frequently observed among women. These patterns reflect sex-specific differences in substrate metabolism: women show higher lipid synthesis rates due to estrogen-driven lipid accumulation, while men exhibit higher protein metabolism. With aging, lipid biosynthesis declines in women, while in men, lipid synthesis increases relative to protein metabolism [50,51,52]. Women also present higher circulating levels of ghrelin (the “hunger hormone”) compared to men, with concentrations declining with age and decreasing testosterone levels [53,54]. Similarly, leptin levels, due to the action of estrogens, are two to three times higher in women than in men, whereas testosterone suppresses leptin synthesis [55,56,57].

Although the study revealed marked differences in dietary habits between students and employees, as well as between men and women, it is essential to examine the underlying determinants of these patterns. Dietary behaviors are shaped not only by knowledge of nutritional guidelines but also by a complex interaction of socioeconomic, cultural, and environmental factors [58]. For instance, students may exhibit higher consumption of fast food and sugar-sweetened beverages due to constrained financial resources, convenience, and the time pressures inherent in academic schedules. Conversely, employees with higher educational attainment and more stable economic conditions are likely to have better access to nutritious foods and greater capacity for meal planning, which may account for their more consistent adherence to nutrient-rich dietary practices. Cultural norms and social expectations further shape eating behaviors [59]. Gender differences observed in our study, such as women prioritizing dietary quality and men favoring high-fat or energy-dense foods, may reflect historical roles, social conditioning, and societal pressures regarding body image and health behaviors. Additionally, living arrangements, family responsibilities, and peer influences are likely to affect meal frequency, portion sizes, and food choices, particularly among students living in shared accommodations or with limited household support [60]. Environmental factors, such as urban versus rural residence, availability of fresh produce, and the density of fast-food outlets, also play a significant role in shaping dietary practices. Furthermore, psychological and lifestyle factors—including stress, workload, and physical activity levels—interact with social and cultural determinants to influence food selection and eating patterns. Recognizing these contextual factors provides a more nuanced understanding of why deviations from dietary recommendations occur and highlights the need for multi-level interventions. Policies and programs aimed at improving nutrition should not only promote knowledge but also address economic accessibility, cultural norms, and environmental constraints to foster sustainable dietary changes across different population groups [61,62,63].

Our study did not reveal cases of alcohol dependence. Most respondents reported only occasional alcohol consumption. However, nicotine use was observed in both groups: 30.76% of students and 10% of employees. Chronic nicotine exposure can disrupt brain development in youth, impair memory and concentration, increase the risk of depression and anxiety, reduce brain volume, and accelerate brain aging, thereby elevating the risk of dementia. Tobacco smoke contains more than 4000 chemical substances, of which at least 40 are proven carcinogens. Among the most harmful compounds are tar, carbon monoxide, pyrene, hydrogen cyanide, toluene, arsenic, urethane, phenol, and benzo[a]pyrene. Nicotine rapidly reaches the brain, where it activates the “reward center” and elevates dopamine levels, producing transient feelings of pleasure, relaxation, and improved mood. However, this effect simultaneously fosters dependence and nicotine craving once the stimulation subsides.

Several limitations should be considered when interpreting the results of this study. First, the research was conducted within a single faculty at Jan Kochanowski University, which limits the generalizability of the findings to other academic institutions, disciplines, or cultural contexts. Second, the sample size was relatively small (39 students and 20 employees), and participation was voluntary, which may have introduced self-selection bias, as individuals experiencing well-being-related issues might have been more motivated to take part. Third, the study did not fully capture the psychological dimensions of well-being, including stress, emotional resilience, and self-efficacy. Consequently, these broader aspects were addressed only theoretically, and future research should include additional validated instruments to assess the full spectrum of well-being. Fourth, the study did not account for potentially confounding factors such as pre-existing medical conditions, medication use, adherence to special diets, or other lifestyle variables that could have influenced the results. Fifth, the cross-sectional and qualitative design of the research limits the ability to establish causal relationships. Sixth, the KomPAN^®^ questionnaire was developed and validated for the adult Polish population. Although this tool had previously been used in studies among students, no separate validation was conducted for this group, which should be considered a potential limitation of this study. Finally, although participant anonymity was ensured and ethical standards were rigorously maintained, the reliance on self-reported data may have introduced reporting bias, particularly regarding sensitive behaviors such as diet, smoking, and other health-related practices.

## 5. Practical Implications

Based on the presented results, the following practical implications for health, education, and health policy can be formulated, which can be grouped into the following thematic areas: (I) targeted nutritional interventions. The results indicate that students consume more vegetables and fruit, but at the same time they are more likely to eat fast food and drink sweetened and energy drinks. This points to the need for targeted nutrition education programs among students that promote healthy alternatives and raise awareness of the consequences of excessive consumption of unhealthy products; (II) occupational health promotion programs. Among employees, the main differences were related to lifestyle, e.g., place of residence and level of education, suggesting that workplace health promotion programs could increase physical activity and improve metabolic health, especially in urban settings; (III) gender-specific health strategies. Gender differences indicate that women are more likely to perform sedentary work and have different eating patterns than men. This suggests a need to tailor intervention strategies to gender, e.g., promoting physical activity among women and educating them about balancing diet and lifestyle; (IV) lifestyle and sleep health promotion. Students devote more time to sleep on weekends, which can be seen as compensation for sleep deprivation on weekdays. Educational programs on sleep hygiene and lifestyle balance could help maintain a regular and healthy circadian rhythm; and (V) holistic approach to well-being. Distinct differences between groups (students vs. employees) and partial differences between genders suggest that interventions should take a holistic approach, considering diet, physical activity, sleep, and environmental factors (number of household members, place of residence).

## 6. Conclusions

The study confirmed significant differences in dietary and lifestyle patterns between students and university employees, as well as between women and men. These findings highlight that well-being within the academic community is shaped by diverse nutritional behaviors and lifestyle choices. Recognizing these differences provides valuable insight for universities to design tailored health promotion strategies and nutritional education programs that address the specific needs of both students and staff.

## Figures and Tables

**Figure 1 nutrients-17-03253-f001:**
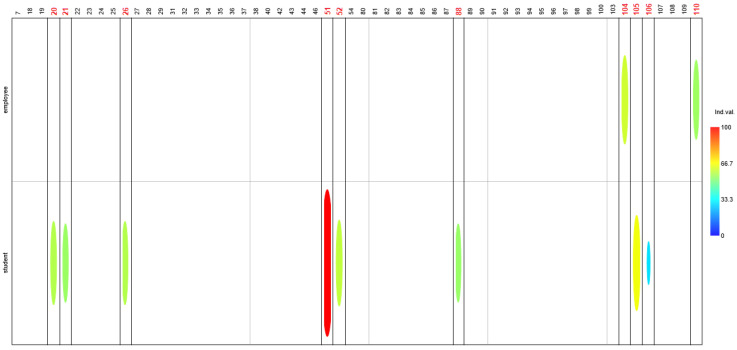
Indicator values differentiating employees and students. Values are expressed as percentages, with color intensity representing their magnitude in the chart. Ordination diagram illustrating statistically significant differences between groups. Question numbers identified as significantly distinguishing a given group are highlighted in red. Only questions with statistically significant differences in responses are shown in the chart.

**Figure 2 nutrients-17-03253-f002:**
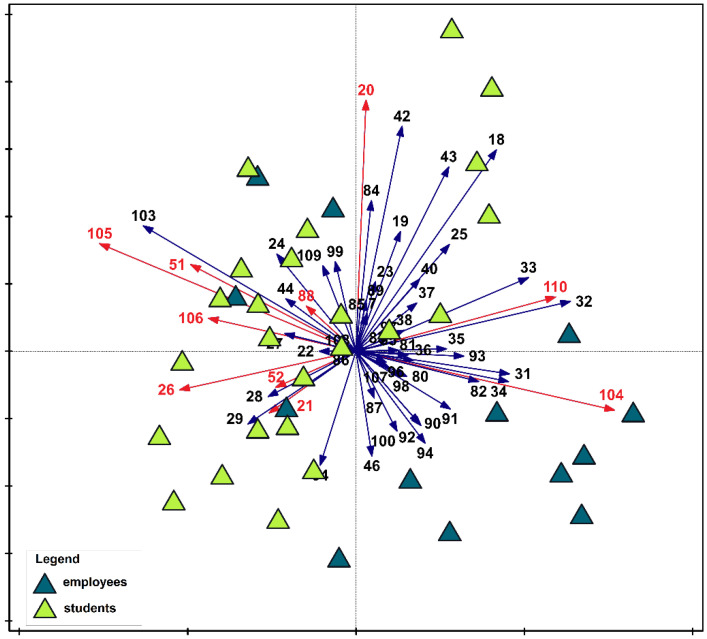
Principal Component Analysis (PCA) of survey data illustrating differences between students and employees. Red vectors indicate question numbers that, according to the indicator value analysis, significantly differentiated the studied groups.

**Figure 3 nutrients-17-03253-f003:**
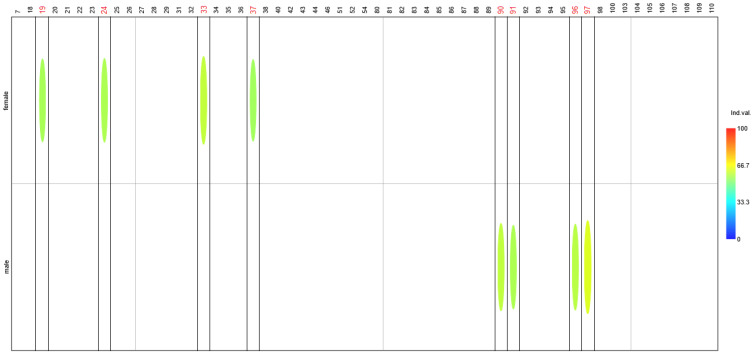
Indicator values differentiating women and men. Values are expressed as percentages, with color intensity representing their magnitude. Question numbers identified as significantly distinguishing a given group are highlighted in red. Only questions with statistically significant differences in responses are shown.

**Figure 4 nutrients-17-03253-f004:**
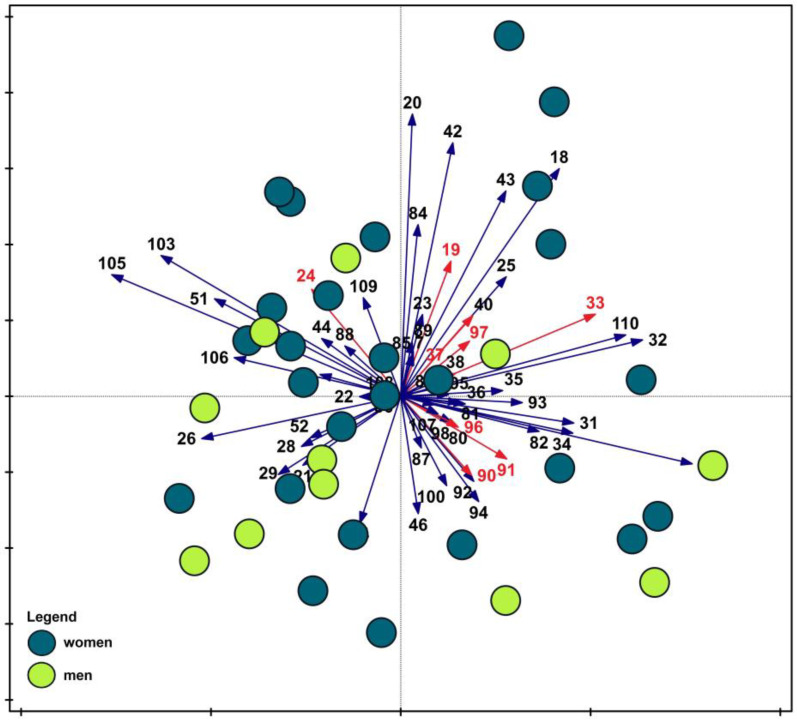
Principal Component Analysis (PCA) of survey data illustrating differences between women and men. Red vectors indicate question numbers that, according to the indicator value analysis, significantly differentiated the studied groups. Legend—Survey question: **7.** Number of meals consumed per day; **18.** Chosen day of the week (from the last week); **19.** Number of meals consumed on the chosen day; **20.** Number of times vegetables or fruit were consumed on the chosen day; **21.** Fast food consumption on the chosen day; **22.** Frequency of consuming white bread; **23.** Frequency of consuming wholemeal/wholegrain bread; **24.** Frequency of consuming white rice, regular pasta, or fine-grain groats; **25.** Frequency of consuming buckwheat groats, oats/oatmeal, or whole-wheat pasta; **26.** Frequency of consuming fast food; **28.** Frequency of consuming butter; **29.** Frequency of consuming lard; **31.** Frequency of consuming milk and milk-based drinks; **32.** Frequency of consuming fermented milk drinks (e.g., yogurts, kefirs); **33.** Frequency of consuming quark/farmer’s cheese; **34.** Frequency of consuming hard/yellow cheese, processed cheese, or blue cheese; **35.** Frequency of consuming processed meats, sausages, or wieners; **36.** Frequency of consuming red meat; **37.** Frequency of consuming white meat; **38.** Frequency of consuming fish; **40.** Frequency of consuming legumes; **42.** Frequency of consuming fruit; **43.** Frequency of consuming vegetables; **44.** Frequency of consuming sweets and confectionery; **46.** Frequency of consuming canned meat; **51.** Frequency of consuming sweetened beverages (carbonated and non-carbonated); **52.** Frequency of consuming energy drinks; **80.** Current diet use; **81.** Type of diet (if applicable); **82.** Duration of diet (if applicable); **83.** Frequency of eating out; **84.** Most frequently consumed type of alcoholic beverage; **85.** Current nicotine product use; **86.** Past nicotine product use; **87.** Sleep duration (weekdays); **88.** Sleep duration (weekends); **89.** Daily screen time; **90.** Self-assessed physical activity (work/school); **91.** Self-assessed physical activity (leisure time); **92.** Self-assessed health status; **93.** Self-assessed nutritional knowledge; **94.** Self-assessed dietary pattern; **95.** Comparison of diet between weekdays and weekends; **98.** Waist circumference (cm); **100.** gender identity; **103.** Year of birth; **105.** Number of people in the household; **106.** Number of minors in the household; **107.** Self-assessed financial situation; **108.** Self-assessed household material status; **109.** Employment status; **110.** Education level.

**Table 1 nutrients-17-03253-t001:** Eligibility criteria for participant selection.

Criteria	Students	Employees (Teaching Staff)
Inclusion	- Full-time student at the Faculty of Exact and Natural Sciences - Aged 19–28 years - Experience of self-reported well-being problems during studies - Willingness to participate in the study	- Full-time research worker at the Faculty of Exact and Natural Sciences - Aged 22–65 years - Experience of self-reported well-being problems during work - Willingness to participate in the study
Exclusion	- Part-time or distance learning students - No experience of well-being-related issues - Refusal to participate	- Part-time employees or administrative staff - No experience of well-being-related issues - Refusal to participate

**Table 2 nutrients-17-03253-t002:** Dietary Habits.

	Group	Subgroup	1	2	3	4	5 and More
Number of meals consumed per day	Students	Male	0	1	6	1	1
		Female	0	6	10	12	2
		Total	0	7	16	13	3
	Employees	Male	0	1	4	1	0
		Female	0	1	7	3	3
		Total	0	2	11	4	3

**Table 3 nutrients-17-03253-t003:** Food Consumption Frequency.

Product Category/Food Item	Group	Subgroup	Never	1–3 Times per Month	Once a Week	Several Times a Week	Once a Day	Several Times a Day
White bread	Students	Male	0	0	1	6	2	0
		Female	0	4	4	11	6	5
		Total	0	4	5	17	8	5
	Employees	Male	1	0	2	3	0	0
		Female	1	1	3	2	2	5
		Total	2	1	5	5	2	5
Wholemeal/wholegrain bread	Students	Male	0	4	1	2	2	0
		Female	9	6	3	10	1	1
		Total	9	10	4	12	3	1
	Employees	Male	2	2	0	1	1	0
		Female	4	5	1	3	1	0
		Total	6	7	1	4	2	0
White rice, regular pasta, or fine-grain groats	Students	Male	0	3	4	1	1	0
		Female	2	2	9	15	1	1
		Total	2	5	14	16	2	1
	Employees	Male	1	3	0	2	0	0
		Female	1	1	6	6	0	0
		Total	2	4	6	8	0	0
Buckwheat groats, oats/oatmeal, whole-wheat pasta, or other coarse-grain groats	Students	Male	1	3	4	0	1	0
		Female	3	16	5	4	2	0
		Total	4	19	9	4	3	0
	Employees	Male	1	4	0	1	0	0
		Female	1	6	4	1	2	0
		Total	2	10	4	2	2	0
Fast food	Students	Male	0	2	4	3	0	0
		Female	1	20	5	4	0	0
		Total	1	22	9	7	0	0
	Employees	Male	2	3	1	0	0	0
		Female	2	10	2	0	0	0
		Total	4	13	3	0	0	0
Fried dishes	Students	Male	0	0	3	4	2	0
		Female	1	4	12	11	2	0
		Total	1	4	15	15	4	0
	Employees	Male	1	4	0	1	1	0
		Female	1	2	5	5	1	0
		Total	2	6	5	6	2	0
Butter	Students	Male	0	0	4	3	2	0
		Female	4	5	4	9	3	5
		Total	4	5	8	12	5	5
	Employees	Male	2	1	0	2	1	0
		Female	1	2	3	3	2	3
		Total	3	3	3	5	3	3
Lard	Students	Male	4	3	1	1	0	0
		Female	23	5	1	1	0	0
		Total	27	8	2	2	0	0
	Employees	Male	4	2	0	0	0	0
		Female	10	3	0	1	0	0
		Total	14	5	0	1	0	0
Milk and milk-based drinks	Students	Male	2	3	0	1	1	2
		Female	0	10	8	5	6	1
		Total	2	13	8	6	7	3
	Employees	Male	1	2	0	2	1	0
		Female	3	2	1	3	3	2
		Total	4	4	1	5	4	2
Fermented milk drinks (yogurts, kefirs)	Students	Male	2	3	1	2	0	1
		Female	2	5	7	9	4	2
		Total	4	8	8	11	4	3
	Employees	Male	0	1	0	2	2	1
		Female	1	2	4	1	6	0
		Total	1	3	4	3	8	1
Quark/Farmer’s cheese	Students	Male	2	5	0	1	1	0
		Female	0	11	6	8	3	2
		Total	2	16	6	9	4	2
	Employees	Male	1	1	2	1	1	0
		Female	1	1	5	4	3	0
		Total	2	2	7	5	4	0
Hard/yellow cheese, processed cheese, blue cheese	Students	Male	0	3	3	1	1	1
		Female	0	4	3	15	5	3
		Total	0	7	6	16	6	4
	Employees	Male	0	2	0	2	2	0
		Female	0	3	3	6	1	1
		Total	0	5	3	8	3	1
Processed meats, sausages, or wieners	Students	Male	0	1	1	4	2	1
		Female	4	8	3	8	7	0
		Total	4	9	4	12	9	1
	Employees	Male	0	0	0	5	1	0
		Female	0	4	3	5	1	1
		Total	0	4	3	10	2	1
Red meat	Students	Male	1	2	2	4	0	0
		Female	9	11	6	4	0	0
		Total	10	13	8	8	0	0
	Employees	Male	1	3	1	1	0	0
		Female	0	7	4	3	0	0
		Total	1	10	5	4	0	0
White meat	Students	Male	0	0	3	4	0	1
		Female	1	3	4	20	2	0
		Total	1	3	7	24	2	1
	Employees	Male	0	3	1	2	0	0
		Female	0	3	4	7	0	0
		Total	0	6	5	9	0	0
Fish	Students	Male	1	7	0	0	0	0
		Female	6	14	8	2	0	0
		Total	7	21	8	2	0	0
	Employees	Male	0	5	1	0	0	0
		Female	1	6	7	0	0	0
		Total	1	11	8	0	0	0
Legumes	Students	Male	3	3	1	1	0	0
		Female	9	13	5	3	0	0
		Total	12	16	6	4	0	0
	Employees	Male	0	5	0	0	1	0
		Female	0	12	2	0	0	0
		Total	0	17	2	0	1	0
Fruit	Students	Male	0	3	0	4	0	1
		Female	0	3	3	13	7	4
		Total	0	6	3	17	7	5
	Employees	Male	0	0	3	1	1	1
		Female	0	0	3	3	6	2
		Total	0	0	6	4	7	3
Vegetables	Students	Male	0	1	0	5	2	0
		Female	0	0	2	15	3	10
		Total	0	1	2	20	5	10
	Employees	Male	0	0	0	5	1	0
		Female	0	0	2	5	5	2
		Total	0	0	2	10	6	2
Sweets and confectionery	Students	Male	0	0	2	4	1	1
		Female	0	4	4	11	6	5
		Total	0	4	6	15	7	6
	Employees	Male	1	0	0	3	2	0
		Female	0	4	1	4	1	4
		Total	1	4	1	7	3	4
Canned meat	Students	Male	6	2	0	0	0	0
		Female	23	7	0	0	0	0
		Total	29	9	0	0	0	0
	Employees	Male	3	3	0	0	0	0
		Female	10	4	0	0	0	0
		Total	13	7	0	0	0	0
Sweetened beverages (carbonated and non-carbonated)	Students	Male	0	2	3	1	1	1
		Female	4	12	4	7	1	2
		Total	4	14	7	8	2	3
	Employees	Male	3	1	1	1	0	0
		Female	3	8	2	1	0	0
		Total	6	9	3	2	0	0
Energy drinks	Students	Male	5	2	2	0	0	0
		Female	13	7	3	6	1	0
		Total	18	9	5	6	1	0
	Employees	Male	3	1	1	1	0	0
		Female	11	3	0	0	0	0
		Total	14	4	1	1	0	0
Alcoholic beverages	Students	Male	2	4	3	0	0	0
		Female	10	19	0	1	0	0
		Total	12	23	3	1	0	0
	Employees	Male	2	2	0	2	0	0
		Female	3	10	1	0	0	0
		Total	5	12	1	2	0	0

**Table 4 nutrients-17-03253-t004:** ParLifestyle and Personal Data.

	Group	Subgroup	Never	1–3 Times Per Month	Once a Week	Several Times a Week	Once a Day	Several Times a Day
Frequency of Eating Out	Students	Male	0	3	3	2	1	0
		Female	0	16	7	7	0	0
		Total	0	19	10	9	1	0
	Employees	Male	1	1	3	1	0	0
		Female	2	6	5	1	0	0
		Total	3	7	8	2	0	0
	**Group**	**Subgroup**	**Less Than 2 h**	**From 2 to Almost 4 h**	**From 4 to Almost 6 h**	**From 6 to Almost 8 h**	**From 8 to Almost 10 h**	**10 h or More**
Daily Screen Time	Students	Male	2	2	1	1	1	2
		Female	3	5	12	5	4	1
		Total	5	7	13	6	5	3
	Employees	Male	1	2	1	2	0	0
		Female	3	4	3	4	0	0
		Total	4	6	4	6	0	0
	**Group**	**Subgroup**		**Insufficient**	**Sufficient**		**Good**	**Very good**
Self-assessed Nutritional Knowledge	Students	Male		1	2		5	1
		Female		2	6		18	4
		Total		3	8		23	5
	Employees	Male		0	3		2	1
		Female		0	4		6	4
		Total		0	7		8	5
	**Group**	**Subgroup**		**Very poor**	**Poor**		**Good**	**Very good**
Self-assessed Dietary Pattern	Students	Male		1	3		5	0
		Female		3	8		18	1
		Total		4	11		23	1
	Employees	Male		1	0		5	0
		Female		0	4		9	1
		Total		1	4		14	1
	**Group**	**Subgroup**		**Essentially no Difference**	**Differs Slightly**		**Differs Significantly**	
Comparison of Diet between Weekdays and Weekends	Students	Male		4	4		0	
		Female		11	15		4	
		Total		15	19		4	
	Employees	Male		1	2		3	
		Female		4	6		3	
		Total		5	8		6	
	**Group**	**Subgroup**		**Worse Than Peers**	**The Same as Peers**		**Better Than Peers**	
Self-assessed Health Status	Students	Male		2	7		0	
		Female		5	22		3	
		Total		7	29		3	
	Employees	Male		1	4		1	
		Female		2	12		0	
		Total		3	16		1	
	**Group**	**Subgroup**		**Low**	**Moderate**		**High**	
Self-assessed Physical Activity (Work/School)	Students	Male		4	3		2	
		Female		20	8		2	
		Total		24	11		4	
	Employees	Male		2	3		1	
		Female		8	6		0	
		Total		10	9		1	
Self-assessed Physical Activity (Leisure Time)	Students	Male		3	3		3	
		Female		14	15		1	
		Total		17	18		4	
	Employees	Male		2	3		1	
		Female		5	8		1	
		Total		7	11		2	
	**Group**	**Subgroup**		**6 or Less Hours/Day**	**More Than 6, but Less Than 9 h/Day**		**9 or More Hours/Day**	
Sleep Duration (Weekdays)	Students	Male		2	7		0	
		Female		16	13		1	
		Total		18	20		1	
	Employees	Male		3	3		0	
		Female		5	9		0	
		Total		8	12		0	
Sleep Duration (Weekends)	Students	Male		1	5		3	
		Female		2	14		13	
		Total		3	19		16	
	Employees	Male		1	4		1	
		Female		0	13		1	
		Total		1	17		2	
	**Group**	**Subgroup**		**No**			**Yes**	
Current Nicotine Product Use	Students	Male		6			3	
		Female		21			9	
		Total		27			12	
	Employees	Male		4			2	
		Female		14			0	
		Total		18			2	
Past Nicotine Product Use	Students	Male		4			5	
		Female		19			11	
		Total		23			16	
	Employees	Male		5			1	
		Female		10			4	
		Total		15			5	

## Data Availability

The original contributions presented in the study are included in the article/Appendix A; further inquiries can be directed to the corresponding author.

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
