# Peer review of "Potential Benefits of Behaviors and Lifestyle for Human Health and Well-Being"

_nutrients, 2025, doi:10.3390/nu17203253_

Round 1
Reviewer 1 Report
Comments and Suggestions for Authors
Dear researchers,
Congratulations on your paper. Here are my comments and/or suggestions:
Introduction:
Lines 33 to 34. Please consider citing an author for this definition of “well-being,” as it is broad from a comprehensive and holistic perspective of people, considering more dimensions (according to the author).
Lines 47 to 48. It is necessary to develop an idea (at the end of the paragraph that ends on line 47) to address the nutritional variable that begins on line 48.
Lines 52 to 57. Are these ideas supported by references 11 and 12?
Lines 59 to 60. An idea needs to be developed (at the end of the paragraph ending on line 59) to address the nutritional variable that begins on line 60.
Lines 60 to 72. Physical activity is mentioned, then sport and exercise are mentioned, and then the benefits of physical activity are revisited. Please consider the order of presentation in this paragraph. It could be structured by mentioning the benefits of physical activity reported in the literature and then pointing out the manifestations of physical activity such as sports and exercise. Another relevant point is that, according to the author, there are recreational sports or physical-sporting activities.
Lines 73 to 80. There is evidence that behavioral tobacco use is adopted because close friends or family members have developed this habit.
Variables were mentioned in the introduction. Are these considered in the objective?
General comments:
It should be noted that this research has presented variables that should be part of the study in general in the introduction, which are not necessarily addressed clearly in the instrument that has been used.
Furthermore, this instrument does not specify further analysis or details of its validation in the population or other characteristics that are relevant to the work presented.
Materials and methods
This section must be included. Please consider “design,” “sample,” “sample selection,” “ethical considerations,” “data analysis,” “procedures,” and other subheadings appropriate for this section of the journal. This is because the introduction mentions the general population without establishing an approximation of the characteristics of this population in general and then mentions university students. There is literature that mentions the characteristics of university administrative staff.
Results
The instrument must be specified. “Products” are mentioned as part of the results. How many participants in the sample are men, women, and of what ages?
Discussion
Your work should highlight strengths, limitations, practical implications, and future lines of research.
Conclusions
Lines 249 to 260: is this information found as part of the objectives or variables addressed in your study? You have presented a lot of information, and it needs to be evidenced in this section as part of the conclusions.
References
Not all articles have a DOI or follow the journal's format.
Author Response
Thank you for your valuable comments. Our detailed point-by-point responses are provided below.

Reviewer 2 Report
Comments and Suggestions for Authors
Dear Authors,
Your research provides a fresh and challenging perspective about the connection between lifestyle and related behaviors with human health.
Please explain extensively how did you select the target group (inclusion/rejecting criteria) and how did you set the number of respondents from both categories (teaching staff and students).
Another aspect, is worth to be clarified, refers to the questionnaire KomPAN, version 2.2 validation before this study.
Please insert Table S1. Dietary Habits and Table S2. Food Consumption Frequency in the text of the paper.
Beyond the strengths of the second applied technique (Principal Component Analysis) as an indirect ordination method, both figures 2 (illustrating differences between students and employees) and figure 4 (illustrating differences between women and men), both on screen and in print are not very easy to decipher for the reader who wants to identify a specific item. Since your analysis revealed statistically significant differences in both comparisons, I presume it is worth trying to improve the supporting iconography.
Most of the 50 selected references are up-to-date and support the topic of the study.
I don’t have any comments regarding the accuracy of English language.
Author Response

(The authors gave the same response as above.)

Reviewer 3 Report
Comments and Suggestions for Authors
This is a very good manuscript that addresses an important topic and provides valuable insights into the relationships between lifestyle, nutrition, and wellbeing.
Some comments/suggestions are the following:
- The study is based on a relatively small sample (n=59), which limits the generalizability of the findings. This should be emphasized as a limitation.
-
All participants were recruited from a single university in Poland. Please clarify that the results may not be representative of other populations or settings.
-
The discussion links results to dietary recommendations but does not sufficiently explore underlying reasons for the observed differences (e.g., socioeconomic or cultural factors).
-
Conclusions – The conclusion is rather general. More specific recommendations (e.g., targeted interventions for students, nutrition education programs, or lifestyle promotion strategies) would strengthen the paper.
Author Response

(The authors gave the same response as above.)

Reviewer 4 Report
Comments and Suggestions for Authors
Evaluation of manuscript nutrients-3893174
I commend you on the work developed in the manuscript titled "Potential Benefits of Behaviors and Lifestyle for Human Health and Wellbeing". The study addresses a topic of great relevance and applicability. However, I would like to offer some specific suggestions to improve the quality of the text.
The current abstract is descriptive in its methodology but does not adequately report the most relevant results. We can observe the following phrase: "The analysis revealed statistically significant differences in both comparisons", but it does not specify what these differences were. Please present more numerical results; the data shown in the supplementary tables are also part of the study.
It is also necessary to review the Objective, as it contradicts what is concluded at the end of the abstract:
“This study examined the dietary habits and lifestyle patterns of employees and students at the Jan Kochanowski University in Kielce (Poland).”
“The primary objective was to assess differences in lifestyle and dietary preferences, first, between university employees and students, and second, between women and men.”
Another point: There is an explicit and non-contextualized mention of "Jan Kochanowski University in Kielce (Poland)" in the abstract and methodology, which makes the study appear as a mere case report of a very specific population without explaining why this particular population is relevant or unique to be studied. If there is no clear justification for specifically studying this population, I recommend removing it from the study.
The introduction of the manuscript does a good job reviewing the general literature on lifestyle, nutrition, and health, clearly establishing the importance of the topic. However, it does not lead the reader to the specific justification for conducting further investigation. Since this is a widely investigated topic, this justification becomes essential. It is noted that there is a jump from the general context to the objective without identifying a knowledge gap or an unresolved research question that this study intends to investigate. When reading the introduction, I wondered: why another study on eating habits? What does this study bring that is new? At the end of the introduction, please add the hypothesis.
Regarding the methods, one of the most critical aspects concerns the sample size. The sample of 59 participants is not based on sample size calculation or statistical power. It is unclear how this number was determined, or whether it has sufficient power to detect significant differences between the compared groups. Additionally, inclusion and exclusion criteria are missing. It is not specified whether factors such as pre-existing health conditions, medication use, special diets, or other factors that could influence the results were considered.
The description of the research instrument is also incomplete. Although the use of the KomPAN questionnaire version 2.2 is mentioned, no information is provided on linguistic or cultural adaptations performed, nor on validation procedures for the specific study population. In which country was it developed? Is it valid for this specific population?
Regarding the statistical analyses, the description is superficial. For the indicator analysis (IndVal), crucial parameters such as the number of permutations performed or the significance level adopted are not specified. In the principal component analysis, essential information on data pre-treatment, such as applied transformations or the type of matrix analyzed (covariance or correlation), is omitted. These omissions prevent adequate understanding of the analytical choices and the evaluation of their appropriateness.
Finally, the absence of any mention of ethical approval of the study by an institutional review board and the obtaining of informed consent from participants constitutes a serious flaw. These are fundamental ethical requirements in research involving human subjects, and their omission raises serious concerns about the conduct of the study. Without these, the study cannot be approved, no matter how relevant it may be.
For the work to be considered for publication, a thorough revision of the methodology section is essential, including: justification of the sample size with statistical power calculation; clear definition of inclusion/exclusion criteria; detailing of the instrument adaptation and validation processes; complete description of statistical procedures with all relevant parameters; and explicit statement of ethical approval and informed consent. These additions are indispensable to ensure the credibility and reproducibility of the study.
The Results section fails to communicate effectively and lacks the statistical rigor to support the findings. The narrative is overly descriptive and qualitative, failing to support the claims of "statistically significant differences" with appropriate numerical data and statistical tests.
The presentation of the data is generic. Statements such as "consumed significantly more" or "significantly more frequent" are insufficient. It is imperative to report the specific values of the statistical comparisons. For each alleged difference, statistical values must be presented, whether significant or not (e.g., p < 0.001), and a measure of effect size (e.g., Cramer's V for chi-square, Cohen's d for t-tests). As it stands, it is impossible to evaluate the findings. Additionally, the exclusive reliance on complex multivariate methods such as IndVal and PCA, without an initial univariate analysis to characterize the sample (such as simple frequencies, means, and standard deviations for the main variables), leaves a gap in the understanding of the basic data.
Figure 1 is hardly comprehensible. An indicator value graph should be self-explanatory. The current legend refers to an extensive and difficult-to-consult list, making the interpretation of the work Herculean. Figure 1 lacks a direct and incorporated legend that clearly identifies what each bar represents (e.g., "Q20: Vegetable consumption", "Q52: Energy drink consumption"). In its current form, the figure does not fulfill its function of communicating information clearly and quickly, becoming an obstacle rather than a tool for understanding.
The Discussion section of the manuscript starts adequately, partially revisiting the objectives and presenting a summary of the main results. However, the transition to contextualization with existing literature could be smoother and more structured.
Regarding the analysis and interpretation of the results, the discussion is theoretically well-founded. The authors make an effort to contrast their findings with the existing literature, especially regarding gender differences in eating behaviors and the relevance of the gut-brain axis. However, if suggestions for a more robust statistical analysis, such as the inclusion of adjusted p-values, effect sizes, or a more disaggregated analysis by subgroups are implemented, it may be necessary to revise the entire discussion or part of it to adapt to the new results.
It is very important to have a paragraph dedicated to the limitations of the study. It is necessary to address limitations such as the small sample size. Finally, I recommend including suggestions for future research, based on the results and limitations found.
Author Response

(The authors gave the same response as above.)

Round 2
Reviewer 1 Report
Comments and Suggestions for Authors
Dear
researchers,
Congratulations on
your work and thank you for the improvements to the paper. I just have a few questions,
please review. After that, it will be ready on my end:
1.- I have questions
about the name of the F1 Supplement. It may cause confusion with Figure 1 already
in the document.
2.- Lines 371 to
373 contain the full article with the DOI. Please check that it matches
the citation. This is repeated in lines 380–382, lines 387–390, and lines 400–
403.
3.- Review the use of “well-being.” or “wellbeing.”
Kind regards.
Author Response
Dear Reviewer,
Thank you for your review and for providing valuable comments. Our detailed point-by-point responses are provided below.
Kind regards.

Reviewer 4 Report
Comments and Suggestions for Authors
Second evaluation of manuscript nutrients-3893174
After a detailed analysis of the response letter and the revised manuscript, it is noticeable that the authors have made substantial improvements, adequately addressing a significant number of my comments. However, they evaded providing complete responses to some of my most critical requests, particularly those related to the methodological core of the study, limiting themselves to stating: "Thank you for your comments."
As requested, the authors addressed and improved the introduction, which now presents a clear rationale for the study, identifies a knowledge gap, and formulates an explicit hypothesis. Similarly, the addition of an extensive and comprehensive section on the study's limitations, addressing issues such as the small sample size and the cross-sectional nature of the research, represents an important improvement.
On the other hand, there are insufficient or evasive responses. Specifically, I had requested that they detail the adaptation and validation processes of the KomPAN® questionnaire for the specific study population. The authors merely cited the original validation article for the general Polish adult population but did not address the central question regarding the instrument's suitability and validity for the specific subgroup of young university students. If this specific validation does not exist, the authors must justify this or include it as a study limitation.
Furthermore, I requested greater statistical rigor in the presentation of the results. The argument that "answers to individual questions are irrelevant" and that only multivariate analyses are justified is questionable. As a current good practice in science, it is recommended that results be supported not only by p-values but also by descriptive statistics and effect size measures. The claim of non-normal data does not invalidate the use of univariate non-parametric tests to corroborate the main findings.
Author Response
Dear Reviewer,
Thank you for your review and for providing valuable comments. Our detailed point-by-point responses are provided below.
Best regards.
